# Hyperforin Enhances Heme Oxygenase-1 Expression Triggering Lipid Peroxidation in BRAF-Mutated Melanoma Cells and Hampers the Expression of Pro-Metastatic Markers

**DOI:** 10.3390/antiox12071369

**Published:** 2023-06-30

**Authors:** Alessia Cardile, Carlotta Passarini, Valentina Zanrè, Alessandra Fiore, Marta Menegazzi

**Affiliations:** Section of Biochemistry, Department of Neuroscience, Biomedicine and Movement Sciences, School of Medicine, University of Verona, Strada Le Grazie, 8, 37134 Verona, Italy; alessia.cardile@univr.it (A.C.);

**Keywords:** heme oxigenase-1, BACH-1, NRF-2, ferroptosis, ferritin, LC3B, CD133, MMP-2, FSP1, uPAR

## Abstract

Hyperforin (HPF) is an acylphloroglucinol compound found abundantly in *Hypericum perforatum* extract which exhibits antidepressant, anti-inflammatory, antimicrobial, and antitumor activities. Our recent study revealed a potent antimelanoma effect of HPF, which hinders melanoma cell proliferation, motility, colony formation, and induces apoptosis. Furthermore, we have identified glutathione peroxidase-4 (GPX-4), a key enzyme involved in cellular protection against iron-induced lipid peroxidation, as one of the molecular targets of HPF. Thus, in three BRAF-mutated melanoma cell lines, we investigated whether iron unbalance and lipid peroxidation may be a part of the molecular mechanisms underlying the antimelanoma activity of HPF. Initially, we focused on heme oxygenase-1 (HO-1), which catalyzes the heme group into CO, biliverdin, and free iron, and observed that HPF treatment triggered the expression of this inducible enzyme. In order to investigate the mechanism involved in HO-1 induction, we verified that HPF downregulates the BTB and CNC homology 1 (BACH-1) transcription factor, an inhibitor of the heme oxygenase 1 (HMOX-1) gene transcription. Remarkably, we observed a partial recovery of cell viability and an increase in the expression of the phosphorylated and active form of retinoblastoma protein when we suppressed the HMOX-1 gene using HMOX-1 siRNA while HPF was present. This suggests that the HO-1 pathway is involved in the cytostatic effect of HPF in melanoma cells. To explore whether lipid peroxidation is induced, we conducted cytofluorimetric analysis and observed a significant increase in the fluorescence of the BODIPY C-11 probe 48 h after HPF administration in all tested melanoma cell lines. To discover the mechanism by which HPF triggers lipid peroxidation, along with the induction of HO-1, we examined the expression of additional proteins associated with iron homeostasis and lipid peroxidation. After HPF administration, we confirmed the downregulation of GPX-4 and observed low expression levels of SLC7A11, a cystine transporter crucial for the glutathione production, and ferritin, able to sequester free iron. A decreased expression level of these proteins can sensitize cells to lipid peroxidation. On the other hand, HPF treatment resulted in increased expression levels of transferrin, which facilitates iron uptake, and LC3B proteins, a molecular marker of autophagy induction. Indeed, ferritin and GPX-4 have been reported to be digested during autophagy. Altogether, these findings suggest that HPF induced lipid peroxidation likely through iron overloading and decreasing the expression of proteins that protect cells from lipid peroxidation. Finally, we examined the expression levels of proteins associated with melanoma cell invasion and metastatic potential. We observed the decreased expression of CD133, octamer-4, tyrosine-kinase receptor AXL, urokinase plasminogen activator receptor, and metalloproteinase-2 following HPF treatment. These findings provide further support for our previous observations, demonstrating the inhibitory effects of HPF on cell motility and colony formation in soft agar, which are both metastasis-related processes in tumor cells.

## 1. Introduction

Heme oxygenase-1 (HO-1) is the rate-limiting enzyme of heme catabolism, converting the heme group into biliverdin, carbon monoxide, and ferrous ion (Fe^2+^) [1]. The activity of HO-1 can elicit different responses in cancer cells, leading them towards either cell death or survival pathways, depending on the type of malignancy and the surrounding environmental conditions. Consequently, numerous discrepancies regarding the pro- and antitumor effects of HO-1 have been reported [1]. The cytoprotective action of HO-1 is mainly mediated by its products, carbon monoxide and biliverdin. The first one is a diffusible gas that acts as a signaling molecule and is involved in angiogenesis and cytokine synthesis [2]. On the other hand, biliverdin has the ability to scavenge hydroxyl radicals and superoxide anions displaying anti-inflammatory and antiapoptotic activities [2]. Iron is an essential cofactor of a large number of key enzymes involved in DNA synthesis and electron transport chain, whose function can support the high proliferation rate of malignant cells [3]. Nevertheless, iron can act as a double-edged sword in tumorigenesis and cancer progression, depending on the balance between the generation of free iron and the capacity to sequester or eliminate it from the cells [4]. In situations where there is an excessive buildup of intracellular free iron and consequent increased production of reactive oxygen species (ROS), the function of HO-1 can switch from being protective to harmful, potentially leading to cell death through a process known as ferroptosis [5]. Ferroptosis is a form of regulated cell death characterized by the iron-dependent accumulation of lipid peroxides and is distinct from other forms of cell death like apoptosis or necrosis [6].

HO-1, being an inducible enzyme, is highly regulated primarily at the transcriptional level. Several redox-sensitive transcription factors, such as the activator protein-1 (AP-1), nuclear factor-κB (NF-κB) and mainly the nuclear factor E2-related factor-2 (NRF-2) have been identified as activators of heme oxygenase-1 gene (HMOX-1) transcription [7]. Additionally, heme-binding transcription factor BTB and CNC homology 1 (BACH-1) can regulate heme levels by modulating HMOX-1 expression through a negative feedback loop [4,8]. Finally, the expression of HO-1 can also be inhibited by microRNAs, which exert their effect through post-transcriptional mechanisms [9,10]. These regulatory mechanisms contribute to the fine-tuning of HO-1 expression in response to different cellular and environmental conditions.

HO-1 protein is typically found at low levels in tissue. Its gene expression can be induced by various factors, including its substrate heme, heavy metals, UV irradiation, ROS, growth factors, and inflammatory cytokines [2]. Given the nature of these inducing agents, it is not surprising that HO-1 is frequently highly expressed in cutaneous melanoma since this skin cancer is promoted by such injuring environmental stimuli [11,12]. The upregulation of HO-1 in melanoma cells may be a cellular response aimed at providing protection against the oxidative stress and inflammation induced by these environmental triggers.

Advanced-stage cutaneous melanoma remains a life-threatening disease, as current therapies have been demonstrated to have limited efficacy in improving patient outcomes [13]. Mutations in the BRAF gene are prevalent in cutaneous melanoma and result in the constitutive activation of mitogen-activated protein kinases (MAPKs), promoting malignant cell growth [14]. While targeted therapy with BRAF inhibitors can delay disease relapse, it offers modest benefits in terms of patients survival due to the emergence of adaptive mechanisms that quickly lead to drug resistance [15]. The hypothesis of HO-1 exerting a pro-tumor role in many cancer types is supported by several authors (reviewed in [16]). In the context of melanoma, HO-1 has been reported to promote cell proliferation through the BRAF-MAPK signaling pathway [11]. Furfaro et al., in their study using primary BRAF-mutated melanoma cells obtained from patients, demonstrated that either the silencing or pharmacological inhibition of HO-1, in combination with a BRAF inhibitor, resulted in a further reduction in cell viability compared to the effect of a drug administered alone [17]. These findings suggest that targeting HO-1 in combination with BRAF inhibitors may have synergistic effects in hampering melanoma cell growth and survival. In addition, HO-1 overexpression has been associated with poor clinical outcomes in uveal melanoma [18]. However, against the HO-1 pro-tumor activities, other studies reported contradictory effects. Was et al. [19] demonstrated that mice with a deficient HMOX-1 genotype developed larger tumors when injected with murine melanoma cells. Conversely, melanomas from wild-type mice exhibited stronger immune cell infiltration, suggesting that HO-1’s antitumor effectiveness is mediated by cells within the tumor micro-environment (TME) [19]. In breast cancer as well, HO-1 may play an antitumor role since its activation has been shown to lead to iron accumulation and cell death via lipid peroxidation [3]. In addition, increased levels of HO-1 protein have been correlated with reduced tumor size and a longer survival time in breast cancer patients [20]. These conflicting findings highlight the complex and context-dependent nature of HO-1 in different types of cancer [5]. It is noteworthy that numerous natural compounds have shown the ability to increase the expression level of HMOX-1/HO-1 (as reviewed in [21]). Indeed, various natural products with anti-inflammatory properties [22,23,24] as well as antitumor activities [25,26,27,28] have been functionally linked to the induction of HO-1. Accordingly, the growth, invasion, colonization, and resistance to therapy of tumors can be countered by phytochemicals through the activation of the HO-1 pathway [29,30,31,32], ultimately resulting in reduced tumor progression.

In our recent study, we demonstrated the remarkable antimelanoma effect of hyperforin (HPF), the main active component of *Hypericum perforatum* extract. We investigated the effects of HPF in three BRAF-mutated melanoma cell lines and observed a decrease in the expression level of glutathione peroxidase-4 (GPX-4) [33]. GPX-4 is a crucial enzyme involved in protecting cells against lipid peroxidation induced by iron. The downregulation of GPX-4 suggests that HPF may disrupt this protective mechanism. Furthermore, we documented a reduction in the malignant phenotype of melanoma cells when exposed to HPF. Specifically, we observed a significant inhibition of cell motility and a decrease in colony formation in soft agar [33]. These findings indicate that HPF has the potential to impede the invasive properties of melanoma cells, which are characteristic features of aggressive and metastatic melanoma.

In the current study, our aim was to explore the connection between decreased antioxidant defense enzymes and iron overload potentially elicited by the dysregulation of proteins controlling iron homeostasis, including HO-1. These events may ultimately contribute to the occurrence of lipid peroxidation.

In three BRAF-mutated melanoma cell lines, we provided evidence that HPF induces HO-1 expression and influences the expression of other proteins involved in iron homeostasis, cell invasion, and metastatic potential.

To further investigate the involvement of HO-1 in the antimelanoma effects of HPF, we silenced the HMOX-1 gene. Through HMOX-1 gene silencing, we were able to demonstrate that the cytostatic activity elicited by HPF in melanoma cells can be partially mediated by HO-1 overexpression.

## 2. Materials and Methods

### 2.1. Cell Cultures

For our study, A375, SK-Mel-28, and FO-1 melanoma cell lines were chosen because they carry the BRAF^V600E^ mutation, which is commonly found in 50–60% of melanoma patients. Melanoma cells harboring the BRAF^V600E^ mutation demonstrate increased aggressiveness and a higher growth rate compared to cells without this mutation. They also exhibit the activation of specific oncogenic pathways in a preferential manner [34]. Additionally, the common feature of these selected melanoma cell lines is their amelanotic nature ([35] and https://www.cellosaurus.org/CVCL_GZ38, Accessed on 3 October 2022). Melanoma cells can exhibit varying levels of pigmentation due to the presence or absence of melanin, which can affect their behavior and response to treatments [36]. These characteristics collectively aim to mitigate the high heterogeneity exhibited by melanoma cell lines. Furthermore, the use of these specific cell lines allows for direct comparisons with previous studies conducted on the same cell lines [33]. This facilitates the interpretation and integration of our present findings with the existing body of knowledge on these cell lines.

Melanoma cell line FO-1 (CRL-12177) (ATCC, Manassas, VA, USA) was cultured in a humidified atmosphere of 5% CO_2_ at 37 °C, in high-glucose Dulbecco’s modified Eagle Medium (DMEM, Gibco, BRL Invitrogen Corp.,Carlsbad, CA, USA). The medium was supplemented with 1% antibiotic antimycotic solution (Gibco, BRL Invitrogen Corp., Carlsbad, CA, USA) and with 10% heat-inactivated fetal bovine serum (FBS; Gibco, BRL Invitrogen Corp., Carlsbad, CA, USA). A375 (CRL 1619) and SK-Mel-28 (HTB-72) melanoma cell lines (ATCC, Manassas, VA, USA) were grown in Roswell Park Memorial Institute 1640 medium (RPMI-1640, Gibco, BRL Invitrogen Corp., Carlsbad, CA, USA) under the previously described conditions. All experiments were conducted using cells that were between 2 and 7 generations old.

### 2.2. Melanoma Cell Treatments

Hyperforin-DCHA (AG-CN2-0008, AdipoGen Life Sciences, Fuellinsdorf, Switzerland) was dissolved in 100% dimethyl sulfoxide (DMSO) at a concentration of 5 mM. Aliquots of the resulting stock solution were stored at −20 °C, protected from light. For the experiments, melanoma cells were treated with HPF concentrations in a range of 2–4 µM and harvested after 24 h for immunoblotting and after 48 h for cell viability assay or lipid peroxidation analysis.

### 2.3. Cell Viability Assay

Melanoma cells were seeded in 96-well plates (12.0 × 10^3^ cells/well for each cell line). The day after, cells were treated with HPF at the concentration of 3 µM, followed by an incubation of 48 h. At the end of the treatment, cells were fixed by adding 25 µL/well of 50% (*w*/*v*) trichloroacetic acid into the culture medium. Plates were then incubated at 4 °C for 1 h, washed 4 times with deionized water (ddH_2_O), and allowed to dry at room temperature (RT). To perform staining, 50 µL/well of 0.04% (*w*/*v*) sulforhodamine B (SRB) sodium salt solution (Sigma-Aldrich, Milan, Italy) was added. After incubating for 1 h at RT, the plates were rinsed with 1% acetic acid and allowed to air dry. SRB stain was then solubilized using a 10 mM Tris-base solution at pH 10.5. Subsequently, the absorbance of the samples was measured at 540 nm using a TECAN NanoQuant Infinite M200 Pro plate reader (Tecan Group Ltd., Männedorf, Switzerland). Five to six replicates were performed for each condition or data point.

### 2.4. Small Interfering RNA Transfection

Small interfering RNA (siRNA) transfection was carried out using ScreenFect^®^siRNA (#S-4001) (ScreenFect GmbH, Eggenstein-Leopoldshafen, Germany) according to the manufacturer’s instructions. The one-step transfection method was employed, which involved simultaneous cell plating and transfection. To prepare the transfection mixture, 0.25 μL of ScreenFect^®^siRNA was vortexed and then diluted in Dilution Buffer to achieve a final volume of 10 μL. For the HMOX-1 siRNA (4390824) and Silencer™ Select Negative Control (4390843) (Life Technologies Corp., Pleasanton, CA, USA), 1 pmol of each siRNA was also diluted in Dilution Buffer to a final volume of 10 µL. The diluted siRNA was then combined with the ScreenFect^®^siRNA dilution and immediately mixed using 10 rapid pipette strokes, without vortexing. The mixture was then left to incubate for 20 min at RT to allow complex formation. Subsequently, 80 µL of freshly detached and resuspended cells (12 × $10^{3}$ cells/well for each cell line: A375, FO-1, SK-Mel-28) were added to the siRNA complexes and mixed using a pipette. Cells and complexes (100 µL) were then seeded into a single well of a 96-well plate. After 24 h of incubation, cells were treated with 3 µM of HPF. Vitality tests were conducted as described previously on 96-well plates, 48 h after treatment. Total protein extracts were obtained from 24-well plates, 24 h after the treatment.

### 2.5. Measurement of Lipid Peroxidation

A375, FO-1, and SK-Mel-28 cells were seeded at a density of 40 × $10^{3}$ cells/well in 24-well plates. After 24 h of incubation at 37 °C, the cells were treated with 2 and 4 µM of HPF. After 48 h, lipid peroxidation was assessed according to the manufacturer’s instructions. Briefly, 5 μM of 4,4-difluoro-5-(4-phenyl-1,3-butadienyl)-4-bora-3a,4a-diaza-s-indacene-3-undecanoic acid, (C11-BODIPY 581/591) (D3861, Thermo Fisher, Waltham, MA, USA) was added to the cells 30 min before the end of the experiment. The mean of fluorescence intensity (MFI) in the samples was measured using flow cytometry (LSR Fortessa with DIVA v9.0 software, BD, Franklin Lakes, NJ, USA) and analyzed using FlowJo 10 software (Tree Star Inc., Ashland, OR, USA).

### 2.6. Total Protein Extracts

Cells were seeded in 24-well plates at a density of 80 × 10^3^ cells/well. Following 24 h, cells were treated with either 2 or 3 µM HPF concentrations or left untreated as a control. Following 24 h of treatment, cells were scraped using warm 1X sample buffer (2% SDS, 10% glycerol, 50 mM Tris-HCl, 1.75% β-mercaptoethanol, and bromophenol blue) and boiled at 99 °C for 10 min. The total protein extracts were then stored at −80 °C until further analysis.

### 2.7. Western Blot Analysis

Protein extracts were subjected to electrophoresis using a 7.5–10% polyacrylamide SDS-PAGE gel. Proteins were subsequently transferred onto a polyvinylidene difluoride membrane (PVDF, Merck-Millipore, Milan, Italy). The membranes were then blocked at RT using a TBST buffer (10 mM Tris-HCl pH 7.5, 100 mM NaCl, 0.1% Tween 20) containing 5% milk for 1 h. Following this, membranes were incubated overnight at 4 °C on a shaker, using a 5% BSA solution containing the primary antibody against HO-1 (A19062), pNRF-2 Ser40 (AP1133), GPX-4 (A11243), SLC7A11 (A2413), FSP1 (A22278), Transferrin (A1448), Ferritin heavy chain (A19544), pRB Ser807/811 (AP0484), (Abclonal, Woburn, MA, USA); LC3B (GTX127375), Cyclin D1 (GTX634347), CD133 (GTX100567), OCT-4 (GTX101497), uPAR (GTX100467), MMP-2 (GTX634832), (Genetex, Alton Parkway, Irvine, CA, USA); AXL (13196-1-AP), BACH-1 (14018-1-AP), (Proteintech, Manchester, UK); GAPDH (2118) (Cell Signaling Technology, Danvers, MA, USA). Subsequently, the membranes were washed three times for 10 min each with TBST buffer. They were then incubated for 1 h with a horseradish-peroxidase-conjugated secondary antibody, either antirabbit or antimouse (Cell Signaling Technology, Danvers, MA, USA). After this incubation, membranes were washed again 3 times for 10 min each with TBST buffer. The expression level of each protein was normalized using GAPDH protein levels, unless otherwise specified. Immuno-detection was performed using an ECL kit (Merck-Millipore, Milan, Italy) and the chemiluminescence signals were visualized using ChemiDoc (Bio-Rad, Hercules, CA, USA) equipment.

### 2.8. Statistics

The results are presented as the mean value ± standard deviation (S.D.). Statistical differences were analyzed using the GraphPad Prism statistical program, and an unpaired, two-tailed Student’s t-test was employed, unless stated otherwise. A p-value less than 0.05 (*), less than 0.01 (**), less than 0.001 (***), or less than 0.0001 (****) was considered to be statistically significant. Each type of experiment was conducted with a minimum of three independent biological replicates. The normal distribution of the data was assessed using the Shapiro–Wilk test.

## 3. Results

### 3.1. Hyperforin Induces HO-1 Protein Expression in A375, SK-Mel-28 and FO-1 Melanoma Cell Lines

Initially, our primary objective was to determine whether HPF induces the expression of HO-1 and to identify the specific transcription factors associated with this induction. After the 24 h treatment of A375, SK-Mel-28, and FO-1 melanoma cells with 2 or 3 µM of HPF, immunoblot analysis revealed an increase in the expression level of HO-1 in all three cell lines (Figure 1). Unexpectedly, our findings indicated that NRF-2, the main transcription factor responsible for the induction of HMOX-1 gene transcription, did not contribute to the higher expression of HO-1. In fact, the phosphorylated and active form of NRF-2 (pNRF-2) exhibited a slight decrease upon HPF treatment (Figure 1). As a result, we investigated the expression level of BACH-1, a transcription factor known for inhibiting HMOX-1 gene expression. Intriguingly, the level of BACH-1 protein expression decreased after HPF treatment (Figure 1), providing an explanation for the HPF-dependent HO-1 upregulation.

### 3.2. HMOX-1 Gene Silencing Partially Reverted the HPF Cytostatic Effect in BRAF-Mutated Melanoma Cell Lines

To silence the HMOX-1 gene, A375, SK-Mel-28, and FO-1 cells were transfected with either HMOX-1 siRNA or a Scramble siRNA as negative control. After 24 h of transfection, the cells were treated with 3 µM of HPF or left untreated. Protein levels were measured via immunoblotting after 24 h, while a sulforhodamine B (SRB) cell viability assay was performed after 48 h. The immunoblots confirmed the high efficiency of HMOX-1 gene silencing in A375 and FO1 cells, leading to the almost complete protein suppression, while the effect in SK-Mel-28 cells was only partial (Figure 2A). This effect was consistent across all independent experimental replicates. Additionally, Figure 2A demonstrates that the decreased BACH-1 expression level induced by HPF was partially restored when the HMOX-1 gene was silenced. Conversely, the expression of pNRF-2 was reduced by HPF treatment regardless of the HMOX-1 expression (Figure 2A).

In A375, SK-Mel-28, and FO-1 cells, we recently reported a 50% reduction in cell viability following a 48 h treatment with 3-4 µM of HPF [33]. In cells transfected with Scramble siRNA, 3 µM of HPF reduced cell viability by 30-50%. However, HMOX-1 silencing samples partially reverted the effect of HPF by significantly restoring cell viability (Figure 2B).

Previously, we demonstrated that HPF can inhibit melanoma cell proliferation by impairing retinoblastoma protein phosphorylation (pRB) and the expression of other cell cycle regulatory proteins, including cyclin D1 [33]. In the current study, immunoblots were performed to verify the decrease in pRB and cyclin D1 expression levels in cells transfected with Scramble siRNA and treated with HPF, compared to untreated cells (Figure 2A). Remarkably, when the HMOX-1 gene was silenced in cells treated with HPF, the expression level of pRB was partially restored in A375 and FO-1 cells, while the level of Cyclin D1 was only restored in the SK-Mel-28 cell line (Figure 2A). These findings suggest that the activation of the HO-1 pathway in melanoma cell lines contributes to the reduction in cell viability elicited by HPF.

### 3.3. HPF Induces Lipid Peroxidation in Melanoma Cells and Alters the Expression of Proteins Involved in Ferroptosis

The upregulation of HO-1 can lead to an accumulation of iron inside cells, which subsequently promotes lipid peroxidation. However, cells have the ability to counteract this by activating antioxidant mechanisms [5].

The cells were treated with either 2 or 4 µM of HPF for 48 h. Subsequently, the cells were labeled with a BODIPY C-11 probe to assess whether HPF treatment can induce lipid peroxidation. The BODIPY C-11 probe has the ability to integrate into cell membranes and emits fluorescence at a wavelength of 590 nm. However, in the presence of lipid peroxidation, the maximal emission of the probe shifts to 510 nm. Flow cytometry was utilized to measure the fluorescence emitted at 510 nm. The results demonstrated a concentration-dependent increase in BODIPY C-11 fluorescence emissions at 510 nm in all cell lines treated with HPF (Figure 3A), suggesting that HPF triggers the lipid peroxidation of cell membranes.

To elucidate the molecular mechanisms responsible for the induction of lipid peroxidation by HPF, we examined the expression levels of various proteins associated with iron homeostasis and ferroptosis. Cells were treated with 3 µM of HPF for 24 h and transfected with either HMOX-1 siRNA or Scramble siRNA.

The measure of GPX-4 protein expression confirmed our previously published findings [33]. Specifically, the expression level of GPX-4 decreased 24 h after the administration of HPF (Figure 3B), suggesting a reduced antioxidant defense capacity of cells against lipid peroxidation.

It should be noted that GPX-4 utilizes glutathione (GSH) to reduce lipo-peroxides into alcohols. Solute carrier family 7 member 11 (SLC7A11) is a subunit of cystine/glutamate antiporter, which plays a crucial role in facilitating the uptake of cystine into the cells, thereby providing an ample supply of cystines for GSH synthesis [37]. In our study, the expression level of SLC7A11 significantly decreased following HPF treatment (Figure 3B), suggesting a limited availability of precursors for GSH synthesis. Apoptosis-inducing factor 2, also known as ferroptosis suppressor protein 1 (FSP1), plays a pivotal role in mitigating lipid peroxidation by reducing oxidized coenzyme Q10 to its reduced form [38]. This enzyme activity prevents the propagation of lipid peroxidation. Importantly, FSP1 operates independently of the SLC7A11/GSH/GPX-4 system to protect the cell membrane [38]. In our experiments, we found that the expression level of FSP1 remained unaffected by HPF treatment (Figure 3B). This suggests that the protective mechanism mediated by FSP1 is not influenced by the treatment.

Transferrin exerts an essential role in maintaining iron homeostasis by controlling iron transport and cellular iron uptake [39]. Immunoblotting analysis revealed the increased expression of transferrin in all cell lines 24 h after treatment with HPF (Figure 3B). Ferritin heavy chain (FTH) is one of the two subunits of ferritin, a protein complex which acts by chelating and storing iron within the cells [40]. The intracellular level of FTH is critical in determining the susceptibility of cells to ferroptosis, as the silencing of the FTH gene can induce ferroptosis [40]. In our experiments, we observed a strong decrease in the expression level of FTH in the presence of HPF (Figure 3B). Additionally, we found that the level of LC3B, a molecular marker of autophagy, was markedly increased in the same samples (Figure 3B and [33]). This suggests that the degradation of ferritin within autophagosomes may be responsible for the depletion of ferritin induced by HPF.

Remarkably, our results showed that changes in the expression of the examined proteins were not reversed by the HMOX-1 silencing, indicating that the effects of HPF on the expression levels of proteins involved in ferroptosis were not directly dependent on HO-1 overexpression (Figure 3B).

### 3.4. HPF Affects the Expression Levels of Certain Proteins Associated with Malignant and Pro-Metastatic Phenotypes in A375, SK-Mel-28, and FO-1 Melanoma Cells

We have recently observed that HPF has the ability to inhibit melanoma cell motility and colony growth in soft agar [33], which are indicative of tumor cells with high metastatic potential. In this study, we investigated the expression levels of specific markers involved in melanoma progression. CD133 is a membrane protein widely recognized as one of the most important markers of cancer stem cells (CSCs) [41]. Immunoblotting data revealed that treatment with 3 µM of HPF for 24 h led to a decrease in CD133 protein expression in melanoma cells (Figure 4). The transfection with HMOX-1 siRNA partially reversed the downregulation of CD133 in A375 and FO-1 cell lines, but it maintained CD133 expression at low levels in the SK-Mel-28 cell line (Figure 4). Another marker of CSCs is octamer-binding transcription factor 4 (OCT-4), which is highly expressed in melanoma cells exhibiting an aggressive phenotype and resistance to anticancer therapy [42]. HPF treatment resulted in a decrease in OCT-4 expression, suggesting that the malignant characteristics of melanoma cells could be decreased by this treatment (Figure 4).Tyrosine-protein kinase receptor UFO (AXL) is a cell surface receptor with tyrosine-kinase activity that can enhance the signaling of growth factors, leading to epithelial-to-mesenchymal transition (EMT) and tumor dissemination [43]. We observed a decrease in AXL expression in cells treated with HPF, as shown in Figure 4. Furthermore, the processes of EMT and melanoma progression are characterized by an elevated expression of the urokinase plasminogen activator receptor (uPAR) [44]. Notably, HPF treatment was found to inhibit uPAR expression, indicating its potential in modulating melanoma progression (Figure 4). Metalloproteinases (MMPs), particularly MMP-2, are known to have crucial roles in cancer cell differentiation, invasion, and metastasis through their ability to degrade extracellular matrix proteins [45]. In our study, we observed a significant decrease in MMP-2 expression 24 h after administering HPF in both A375 and FO-1 cell lines, whereas SK-Mel-28 expressed MMP-2 at a lower level without any change after HPF treatment. Notably, when FO-1 cells were transfected with HMOX-1 siRNA, the expression of MMP-2 newly increased once again (Figure 4). In summary, our findings demonstrate a widespread decrease in the expression levels of melanoma progression markers upon HPF treatment. This effect provides a possible explanation for the previously observed attenuation of cell mobility and colony formation in soft agar within the same melanoma cell lines.

## 4. Discussion

Hyperforin, as the primary active component of *Hypericum perforatum* extract, exhibits various biological activities. Among them, HPF has been found to exert several antitumor effects, specifically in melanoma cells. These effects include inhibiting cell growth and colony formation in soft agar, slowing down cellular motility, and promoting apoptosis [33,46]. Notably, our previous study has shown that when BRAF-mutated melanoma cell lines are treated with low micromolar concentrations of HPF, among many other proteins, they experience a depletion of GPX-4 [33]. GPX-4 is a crucial enzyme that utilizes GSH to counteract iron-induced lipid peroxidation in cell membranes. These findings have prompted further investigation into the potential role of iron metabolism in the molecular mechanisms underlying the antimelanoma activity of HPF. HO-1 is an enzyme that catalyzes the heme group to release ferrous ion as one of its products [47]. In our study, we found a significant increase in the expression of HO-1 protein in A375, SK-Mel-28, and FO-1 melanoma cells treated with 2 and 3 µM of HPF for 24 h (Figure 1). These results partially align with a recent report that demonstrated the high expression of HO-1 in WM115 and WM266-4 melanoma cell lines treated with *Hypericum perforatum* extract. However, the response to HPF treatment showed variability in that study [46].

Notably, HO-1 is known as an inducible enzyme, and its expression level is tightly regulated [4,7,8,9,10]. The promoter region of the HMOX-1 gene contains binding sites for both NF-ҝB and AP-1 transcription factors [48]. In our previous study with the same cell lines, we observed a decrease in the phosphorylated and active form of the NF-ҝB-P65 subunit following HPF treatment [33]. This finding means it is unlikely that NF-ҝB plays a significant role in the activation of HO-1. Regarding the AP-1 transcription factor, our previous study demonstrated that two components of the AP-1 complex associated with tumor progression, namely Fos-related antigen 1 (FRA-1) and phospho-cJun, exhibited a time-dependent expression pattern following HPF administration. Specifically, their expression increased from 30 min until 6 h after HPF treatment but decreased at 24 h and at later time points [33]. These findings suggest that the AP-1 complex may contribute to the induction of HO-1 by HPF, at least during the early stages of treatment. The NRF-2 transcription factor is a key regulator of the HMOX-1 gene and is often overexpressed due to the action of natural antioxidant compounds [49]. However, in our study, we observed a slight decrease in the phosphorylated and active form of NRF2 in all melanoma cell lines following HPF treatment (Figure 1). This finding suggests that the HPF-induced overexpression of HO-1 is not dependent on NRF-2 activity. It is worth noting that NRF-2 upregulation has been associated with promoting the oncogenic potential of cancer cells through the overexpression of several antioxidant enzymes [49,50,51]. Therefore, the downregulation of NRF2 in melanoma cells upon HPF treatment may contribute to the antitumor effects of HPF. The transcription factor BACH-1, similar to NRF2, can bind to antioxidant-responsive elements in response to changes in redox states [4,8]. However, BACH-1 functions primarily as a transcriptional repressor for several antioxidant genes, including HMOX-1 [52]. Conversely, BACH1 has been shown to activate the transcription of glycolytic enzymes, leading to increased glucose uptake, glycolysis rates, and lactate secretion, thereby promoting the glycolysis-dependent metastasis of both mouse and human lung cancer cells [52]. Indeed, the decrease in BACH-1 expression observed upon HPF administration (Figure 1) supports the hypothesis that HPF may induce HO-1 expression by blocking the inhibitory effect of BACH-1 on HMOX-1 gene promoter. Furthermore, the reduction in the expression levels of some glycolytic enzymes obtained 24 h after HPF administration, as reported in our previous report [33], can be endorsed by the decreased expression of BACH-1. This effect potentially impairs the glycolytic metabolism associated with tumor progression and metastasis.

It has been reported that HO-1 induction can promote cell proliferation through the BRAF-MAPK signaling pathway in melanoma [11]. Additionally, the overexpression of HO-1 has been shown to increase the proliferative rate of murine melanoma cells B16-F10 and enhance the aggressiveness of melanoma cells in mice [53], although the overexpression of HO-1 during clonal growth induction in vitro and in vivo can play an antitumorigenic role [54]. To further investigate the role of HO-1 in the pleiotropic antitumor effects elicited by HPF, we assessed melanoma cell viability after silencing the HO-1 gene using siHMOX-1 cell transfection, compared to cells transfected with scrambled siRNA. Our results demonstrated that siHMOX-1 transfection effectively reduced HO-1 protein expression in A375 and FO-1 cell lines and induced a partial but significant reduction in SK-Mel-28 cells.

We observed a significant decrease in the expression level of BACH-1 when cells were treated with HPF and transfected with scrambled siRNA. Interestingly, when the HMOX-1 gene was silenced, there was a slight increase in BACH-1 expression (Figure 2A), suggesting a negative correlation between BACH-1 and HO-1 protein levels. This negative correlation is modulated by BACH-1, exerting inhibitory activity on HMOX-1 gene transcription, which contributes to the downregulation of HO-1 expression. On the other hand, when the heme group is present intracellularly, it can bind to BACH-1 and promote its degradation. HO-1, being an enzyme involved in heme degradation, can impact the intracellular level of heme, thus influencing the stability and expression of BACH-1 [4]. The interplay between BACH-1, HO-1, and heme levels highlights the complex regulatory mechanisms involved in the modulation of HO-1 expression.

In Figure 2B, we observe a slight but significant increase in cell viability when the HMOX-1 gene is silenced in A375, SK-Mel-28, and FO-1 melanoma cells. This finding suggests that the inhibitory effect of HPF on melanoma cell viability may be mediated, at least in part, through the activation of the HO-1 pathway. To further support this hypothesis, we examined the expression level of phosphorylated and active forms of retinoblastoma protein. pRB is a key regulator of the cell cycle, and its phosphorylation status is closely linked to cell proliferation. We found that pRB level decreased with HPF treatment in cells transfected with scrambled siRNA. However, in cells transfected with HMOX-1 siRNA, we observed a slight increase in pRB level (Figure 2A). The expression level of cyclin D1 was reduced after HPF treatment, although its protein amount was only partially recovered in SK-Mel-28 cells when the HMOX-1 gene was silenced (Figure 2A).

Overall, the upregulation of HO-1 induced by HPF could support the antiproliferative effect of this phytocompound by partially hindering RB phosphorylation or cyclin D1 expression, although the specific molecular mechanism was not investigated in this study. Remarkably, our data suggest that HO-1 activity could display antitumor effects depending on the cellular context, as observed in this study where it was conditioned by the severe alterations induced by HPF treatment on the expression levels of crucial proteins involved in cell cycle regulation.

HO-1 upregulation can induce iron overload, which in turn can result in lipid peroxidation and ferroptosis [4,5,47]. However, cells display a cluster of enzymes to contrast the dangerous effects elicited by excessive intracellular iron. Therefore, we directly measured lipid peroxidation through flow cytometric analysis by staining cells with a fluorescent probe (BODIPY C-11). After 48 h of HPF treatment, cells loaded with BODIPY C-11 showed higher levels of fluorescence in comparison with untreated cells (Figure 3A), confirming that cells were subjected to lipid peroxidation. However, to investigate the molecular mechanisms underlying the induction of lipid peroxidation, we analyzed the expression level of additional proteins involved in iron metabolism and ferroptosis.

A key enzyme that protects cells from lipid peroxidation is GPX-4, a target of class II ferroptosis inducers like RSL3 [55,56]. Consistent with previous findings, we observed a significative reduction in GPX-4 expression levels 24 h after HPF administration (Figure 3B). Another target of class I ferroptosis inducers like erastin is the Xc- system, which is a cystine/glutamate antiporter that achieves cystine uptake, the necessary source of cysteines for GSH synthesis [37]. Indeed, erastin sensitizes cancer cells to ferroptosis via GSH starvation by inhibiting the SCL7A11 subunit of the Xc- system [57]. In our experiments, we found that HPF was able to reduce SLC7A11 expression in A375 and FO-1 melanoma cells, but not in the SK-Mel-28 cell line (Figure 3B). Given that GPX-4 consumes GSH to reduce lipid peroxides, the limited availability of both GPX-4 and GSH can promote lipid peroxidation.

Cellular defense mechanisms also involve the FSP1 enzyme, which catalyze the conversion of coenzyme Q from ubiquinone to ubiquinol, thereby capturing lipid peroxyl radicals [58,59]. FSP1 acts as a GSH/GPX-4-independent ferroptosis suppressor [58,59]. In melanoma cells treated with HPF, the FSP1 expression level was unchanged (Figure 3B), suggesting that this antioxidant-protective enzyme was not affected by HPF administration. Given that lipid peroxidation was observed in all cell lines examined, it is evident that, in the presence of HPF, this protective system may not be fully effective. Additional investigations will be required in future studies to better understand the factors contributing this limited effectiveness.

Transferrin is the principal iron carrier of biological fluids, and it can be internalized into cells along with its iron content through the transferrin receptor [60]. Recent studies have highlighted that high levels of intracellular transferrin are associated with increased susceptibility to ferroptosis, since the silencing of the transferrin gene can attenuate this specific form of cell death [61,62]. Immunoblots revealed higher expression levels of transferrin in melanoma cells treated with HPF (Figure 3B), indicating that iron uptake could be increased.

However, to counteract the potentially harmful effects of high levels of intracellular free iron, cells rely on ferritin, which is able to sequester iron excess. Autophagy-dependent ferroptosis involves the autophagic degradation of proteins associated with ferroptosis, including GPX-4 and ferritin [63]. Notably, autophagy can trigger ferroptosis by degrading ferritin in both fibroblasts and cancer cells [64]. In cells treated with HPF, we detected very low levels of ferritin concomitantly with a strong increase in the expression of the autophagy marker LC3B (Figure 3B).

Based on our findings, it can be summarized that the lipid peroxidation triggered by HPF is primarily attributed to the reduction in expression levels of GPX-4, ferritin, and SLC7A11, potentially facilitated by autophagic degradation. This phenomenon is likely intensified by the concomitant increase in iron uptake through transferrin.

Remarkably, the silencing of the HO-1 gene does not appear to have an impact on the expression levels of the investigated proteins linked to lipid peroxidation. However, the HPF-induced upregulation of HO-1 may contribute to reinforcing lipid peroxidation by increasing the levels of free iron that cannot be effectively neutralized due to ferritin depletion.

There is indeed substantial evidence supporting the significant involvement in melanoma progression of the resistance to the ferroptosis pathway [65]. Our findings highlight the complexity of the cellular response to HPF and suggest the involvement of several alterations in protein expression targeted by HPF. The combination of these alterations, leading to iron unbalance and lipid peroxidation, may sensitize melanoma cells to cell death through ferroptosis.

In our recent study, we demonstrated a decrease in the malignant phenotype of melanoma cells treated with HPF. We observed a reduction in cell motility and impaired colony formation in soft agar [33]. Building upon these findings, our current research focused on investigating the expression of proteins involved in cell invasion and metastatic potential in the same melanoma cell lines treated with this phytocompound.

The expression of CD133 has been recognized as crucial for human metastatic melanoma cells [41]. The silencing of the CD133 gene has been shown to inhibit growth and migratory capability [41]. In our immunoblot analysis, we observed a marked reduction in CD133 expression levels in all melanoma cell lines transfected with scrambled siRNA and treated with HPF (Figure 4). Notably, the treatment with HPF in A375 and FO-1 cells transfected with HMOX-1 siRNA resulted in a partial restoration of CD133 expression levels (Figure 4), suggesting that the HO-1 pathway may be involved in the downregulation of this oncoprotein induced by the phytocompound.

Melanoma cells expressing the OCT-4 protein exhibited a more aggressive phenotype [42]. Treatment with HPF can inhibit OCT-4 expression, regardless of whether the HMOX-1 gene is silenced or not (Figure 4). AXL, a tyrosine-kinase receptor, is highly expressed in malignant melanoma. Increased AXL signaling has been associated with drug resistance, particularly in the case of BRAF or MEK inhibitors [66]. Consequently, a combined therapy involving a BRAF/MEK inhibitor along with an AXL inhibitor is being considered for enrollment in a clinical trial [67]. In vitro studies have shown that AXL silencing via siRNA or pharmacological inhibition dramatically impedes the migration and invasion of melanoma cells [68]. The inhibition of uPAR expression using a specific uPAR antisense oligonucleotide has been found to inhibit cell invasion, angiogenesis, metastasis, and MMP expression [44,69]. MMPs are enzymes that can degrade the extracellular matrix, thereby promoting invasion. Specifically, the expression level of MMP-2 has been shown to be associated with poor prognosis in patients carrying BRAF-mutated melanoma [45].

In all cell lines, treatment with HPF resulted in a decrease in the expression levels of AXL, uPAR, and MMP2 (Figure 4). Notably, when comparing SK-Mel-28 and FO-1 melanoma cells to A375 cells, which are reported to exhibit higher metastatic potential, enhanced expression levels of AXL, uPAR, and MMP2 were observed in A375 cells (Figure 4).

Except for CD133 in A375 and FO-1 cells and MMP-2 in the FO-1 cell line, the silencing of the HMOX-1 gene did not reverse the protein downregulation induced by HPF treatment. This suggests that each cell line responds to the treatment through the activation of different pathways and signaling molecules.

## 5. Conclusions

In conclusion, this study elucidated the effects of HPF treatment on melanoma cells and its relationship with HO-1 expression, lipid peroxidation, iron homeostasis, and melanoma progression markers. The results demonstrated that HPF treatment induced HO-1 expression in melanoma cells, which was not dependent on the NRF-2 transcription factor but was potentially regulated by its transcriptional repressor BACH-1. HPF treatment also triggered lipid peroxidation, accompanied by a decrease in GPX-4 and SLC7A11 expression, suggesting reduced antioxidant capacity and limited availability of precursors for GSH synthesis. Furthermore, HPF treatment could lead to an augmentation in the intracellular free iron due to increased transferrin and HO-1 expression and decreased ferritin. The study also revealed that HPF treatment downregulated melanoma progression markers, such as CD133, OCT-4, AXL, uPAR, and MMP-2, which are associated with tumor growth, metastasis, and EMT. The silencing of the HMOX-1 gene partially reversed the effects of HPF on cell viability, pRB, BACH-1, and CD133 expression. Overall, these findings suggest that HPF exerts its antimelanoma effects through pleiotropic effects, including the induction of HO-1, triggering lipid peroxidation, disrupting iron homeostasis, and attenuating melanoma progression markers.

This study has certain limitations:To address the high heterogenicity of melanoma cell lines, we specifically selected three highly aggressive cell lines harboring the BRAF^V600E^ mutation, which exhibit the activation of several onco-proteins [34]. These cell lines are also homogenous due to their amelanotic nature. Since the presence or absence of melanin can affect cell behavior and response to therapy [36], we cannot exclude the concept that other cell lines with different characteristics may exhibit varying responses.Not all proteins involved in iron metabolism and lipid peroxidation were investigated. Further research is required to elucidate other potential intracellular targets of HPF associated with iron homeostasis and/or lipid peroxidation. Specifically, certain enzymes involved in both the coenzyme Q biosynthesis and the metabolism of mono- and poly-unsaturated fatty acids may play a significant role.Our experiments were exclusively conducted in vitro, which may not fully reflect the complex interactions and dynamics present in an in vivo setting.During tumor progression, melanoma cells generate neurotransmitters that regulate TME cells and impact tumor homeostasis [70]. Given that HPF can modulate multiple neurotransmitter signaling pathways, it is crucial to conduct in vitro and in vivo studies to further comprehend this complex interplay.

To sum up, the recent review [71] highlights that HPF exhibits diverse molecular mechanisms that have a broad impact on various types of cancer. Since the finding of this study have attested that HPF’s antitumor effect on melanoma cells also involves iron-dependent lipid peroxidation, exploring this effect in different type of cancer cells would be intriguing.

## Figures and Tables

**Figure 1 antioxidants-12-01369-f001:**
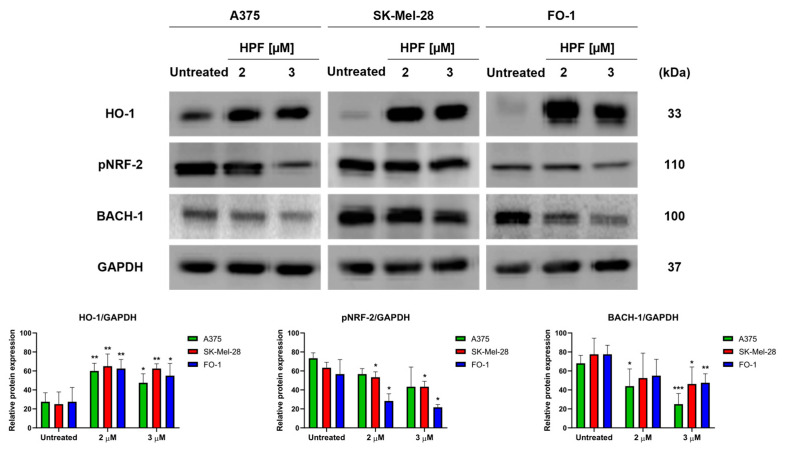
Hyperforin induces heme oxygenase-1 (HO-1) expression in melanoma cell lines. A375, SK-Mel-28, and FO-1 melanoma cells were treated with 2 or 3 µM of hyperforin (HPF) for 24 h. (Top) Representative immunoblots show the expression levels of HO-1, phosphorylated form of nuclear factor E2-related factor-2 (pNRF-2), heme-binding transcription factor BTB and CNC homology 1 (BACH-1), and glyceraldehyde-3-P dehydrogenase (GAPDH) after 24 h of treatment with 2 or 3 µM of HPF. (Bottom) Histograms represent the mean values ± S.D. of protein expression levels, as measured via densitometry in three independent experiments and normalized with GAPDH expression. Statistical analysis was performed using Student’s *t*-test for treated samples compared to the respective untreated samples after data normalization. Significance levels are denoted as follows: * *p* < 0.05; ** *p* < 0.01; *** *p* < 0.001.

**Figure 2 antioxidants-12-01369-f002:**
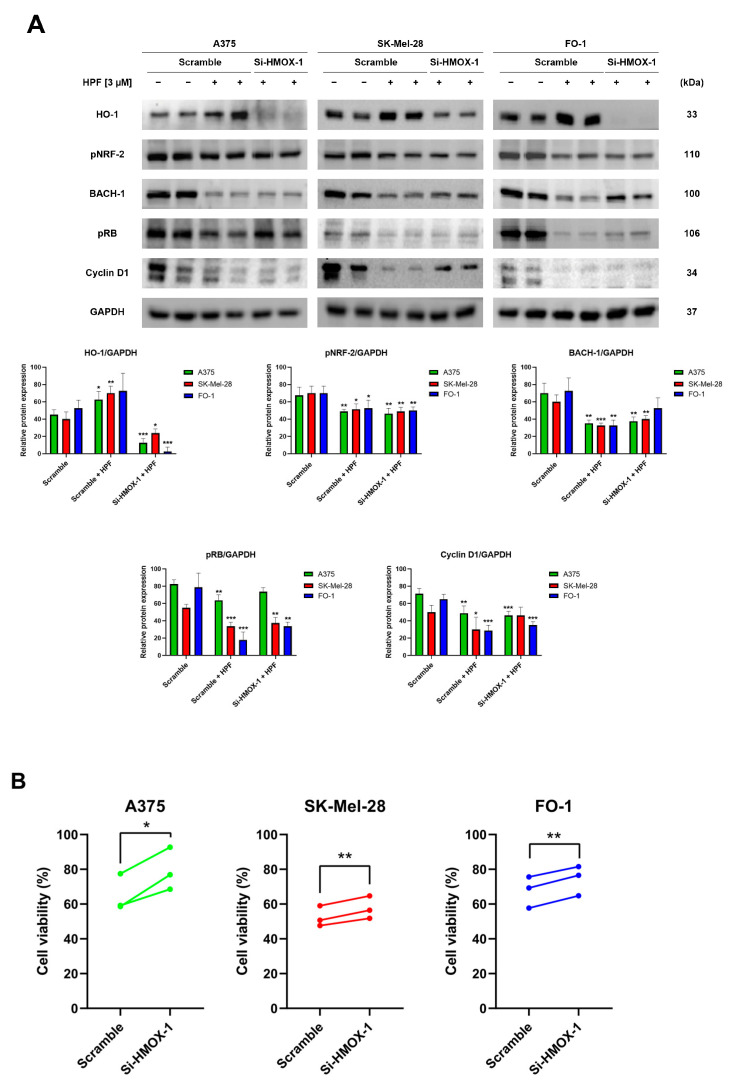
Hyperforin affects cell viability and the expression of cell-cycle-regulating proteins in cells silenced using HMOX-1 siRNA. A375, SK-Mel-28, and FO-1 melanoma cells were transfected with either HMOX-1 siRNA or a scrambled siRNA. After 24 h of transfection, cells were treated with or without 3 µM of HPF. (**A**) Representative immunoblots show the expression levels of HO-1, pNRF-2, BACH-1, phosphorylated form of retinoblastoma protein (pRB), Cyclin D1, and GAPDH after 24 h of treatment with 3 µM of HPF. The histograms display the mean values ± S.D. of protein expression levels, measured via densitometry from three independent experiments and normalized to GAPDH expression. All comparisons were performed against each respective untreated sample after data normalization. (**B**) The SRB cell viability assay was performed on A375 (green), SK-Mel-28 (red), and FO-1 (blue) cells. Before treatment with 3 µM of HPF for 48 h, cells were transfected with either HMOX-1 siRNA or a scrambled siRNA. Statistical comparisons were conducted using Student’s *t*-test for unpaired samples for immunoblots and paired samples for SRB cell viability assay. * *p* < 0.05; ** *p* < 0.01; *** *p* < 0.001.

**Figure 3 antioxidants-12-01369-f003:**
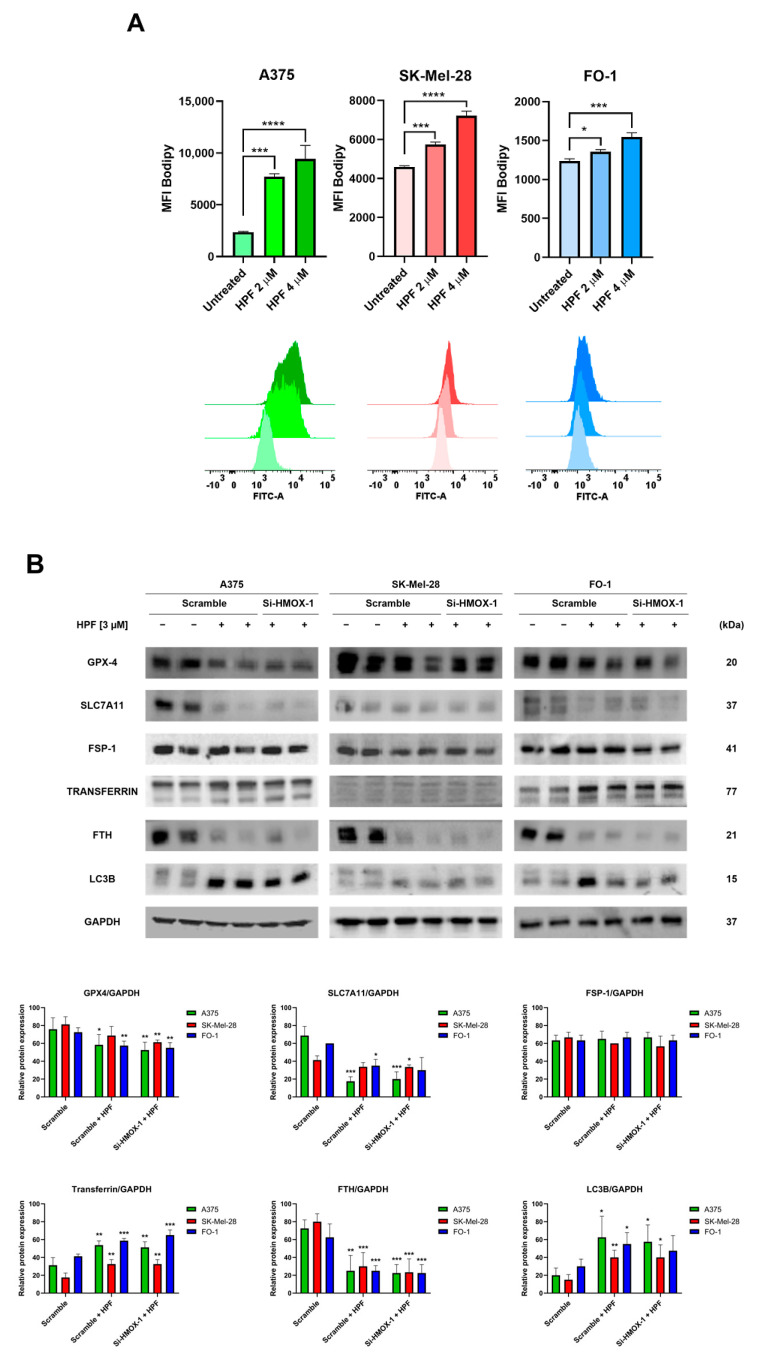
Hyperforin induces lipid peroxidation and affects the expression of proteins associated with iron homeostasis. (**A**) A375, SK-Mel-28, and FO-1 melanoma cells were treated with or without HPF for 48 h, followed by labeling with BODIPY C-11 probe. Fluorescence was measured using cytofluorimetric analysis. (**B**) Cells were transfected with either HMOX-1 siRNA or scrambled siRNA. After 24 h of transfection, cells were treated with or without 3 µM of HPF. Representative immunoblots showed the expression levels of glutathione peroxidase-4 (GPX-4), solute carrier family 7 member 11 (SLC7A11), ferroptosis suppressor protein 1 (FSP1), transferrin, ferritin heavy chain (FTH), light chain 3 B (LC3B), and GAPDH. Histograms represent the mean values ± S.D. of protein expression levels measured via densitometry from three independent experiments and normalized to GAPDH expression. Student’s *t*-test was used for unpaired samples to each untreated sample after data normalization. * *p* < 0.05; ** *p* < 0.01; *** *p* < 0.001, **** *p* < 0.0001.

**Figure 4 antioxidants-12-01369-f004:**
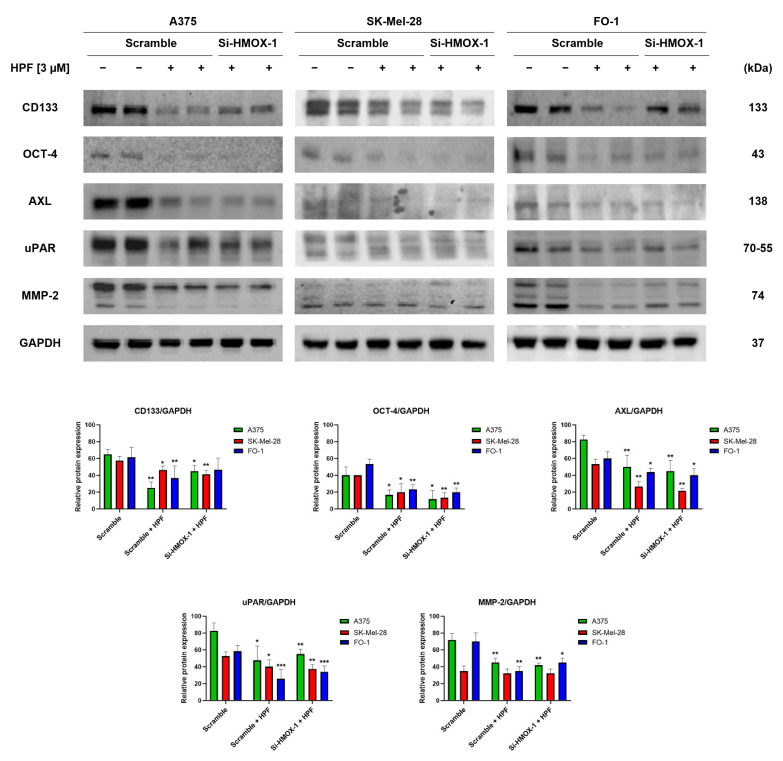
Hyperforin affects the expression of proteins involved in tumor invasion and metastatic potential. Cell were transfected with either HMOX-1 siRNA or a scrambled siRNA. After 24 h of transfection, the cells were treated with or without 3 µM of HPF. Representative immunoblots show the expression level of CD133 antigen, octamer-4 (OCT-4), tyrosine-protein kinase receptor UFO (AXL), urokinase plasminogen activator receptor (uPAR), metalloproteinase-2 (MMP-2), and GAPDH as a loading control. The histograms represent the mean values ± S.D. of protein expression levels, which were measured via densitometry and derived from three independent experiments. Statistical comparisons were conducted using Student’s *t*-test for unpaired samples after data normalization with GAPDH expression. * *p* < 0.05; ** *p* < 0.01; *** *p* < 0.001.

## Data Availability

Not applicable.

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
