# Peer review of "Hyperforin Enhances Heme Oxygenase-1 Expression Triggering Lipid Peroxidation in BRAF-Mutated Melanoma Cells and Hampers the Expression of Pro-Metastatic Markers"

_antioxidants, 2023, doi:10.3390/antiox12071369_

Round 1

Reviewer 1 Report

This is an interesting paper which, however, would benefit from some revisions.

The topic is of interest. Methodology for the most part is adequately described. The data appears to be properly collected and analyzed.

My critique is as follows.

Better description of original properties of melanoma lines would be expected. For example, melanin pigmentation and stimulation of melanogenesis can affect melanoma behavior and therapy (Frontiers in Oncology 2022;12. DOI: 10.3389/fonc.2022.842496). The phenotype as relates to melanogenic behavior in melanoma lines is not described. Some are amelanotic (A375). Media used, DMEM, stimulate melanogenesis  and may have negative selection pressure during serial passaging. These issues could be mentioned in limitations and role of melanin pigment also mentioned as listed above.

Also lack of in vivo experiments should be mentioned in limitations

Also the readers would be interested in brief mentioning that melanoma can affect body mechanisms through diverse mechanisms as discussed recently (Trends Neurosci 46: 263-275, 2023. https://doi.org/10.1016/j.tins.2023.01.003)

Author Response

Answer to Reviewer 1

We thank the reviewer for the valuable suggestions he/she addressed in order to improve the present manuscript.

Better description of original properties of melanoma lines would be expected. For example, melanin pigmentation and stimulation of melanogenesis can affect melanoma behavior and therapy (Frontiers in Oncology 2022;12. DOI: 10.3389/fonc.2022.842496). The phenotype as relates to melanogenic behavior in melanoma lines is not described. Some are amelanotic (A375). Media used, DMEM, stimulate melanogenesis and may have negative selection pressure during serial passaging. These issues could be mentioned in limitations and role of melanin pigment also mentioned as listed above.

Answer: we appreciate the reviewer’s feedback, and agree with his/her suggestion. In the original version of the manuscript, we inadvertently omitted a description of the genotype and phenotype of the cell lines used. Therefore, we have made the necessary revisions in the Materials and Methods Section. The following sentences have been added to address this oversight:

"For our study, A375, SK-Mel-28, and FO-1 melanoma cell lines were chosen because they carry the BRAFV600E mutation, which is commonly found in 50-60% of melanoma patients. Melanoma cells harboring the BRAFV600E mutation demonstrate increased aggressiveness and a higher growth rate compared to cells without this mutation. They also exhibit the activation of specific oncogenic pathways in a preferential manner [34]. Additionally, the common feature of these selected melanoma cell lines is their amelanotic nature ([35], and https://www.cellosaurus.org/CVCL_GZ38). Melanoma cells can exhibit varying levels of pigmentation due to the presence or absence of melanin, which can affect their behavior and response to treatments [36]. These characteristics collectively aim to mitigate the high heterogeneity exhibited by melanoma cell lines. Furthermore, the use of these specific cell lines allows for direct comparisons with previous studies conducted on the same cell lines [33]. This facilitates the interpretation and integration of our present findings with the existing body of knowledge on these cell lines.

All experiments were conducted using cells that were between 2 and 7 generations old."

Regarding the potential effect of the cell medium used, we mistakenly indicated DMEM as medium used for both A375 and FO-1 cells in the original version. However, we have now corrected this error in the revised version. Indeed, we used DMEM only for FO-1 cell line. It is worth noting that in the past, we have also used DMEM medium for A375 cells. Based on our experience, we have not observed any significant differences in the cell phenotype when using either RPMI or DMEM medium, possibly because the cells we used were between 2 to 7 generation old. However, we acknowledge that this variable should be considered in future studies.

 Also lack of in vivo experiments should be mentioned in limitations

Also the readers would be interested in brief mentioning that melanoma can affect body mechanisms through diverse mechanisms as discussed recently (Trends Neurosci 46: 263-275, 2023. https://doi.org/10.1016/j.tins.2023.01.003)

Answer: we agree with the reviewer. In the Conclusion Section of the revised manuscript, we added the following sentences regarding the limitations of this study:

"This study has certain limitations:

  1. to address the high heterogenicity of melanoma cell lines, we specifically selected three highly aggressive cell lines harboring the BRAFV600E mutation, which exhibit the activation of several onco-proteins [34]. These cell lines are also homogenous for their amelanotic nature. Since the presence or absence of melanin can affect cell behavior and response to therapy [36], we cannot exclude that others cell lines with different characteristics may exhibit varying responses;
  2. not all proteins involved in iron metabolism and lipid peroxidation were investigated. Further research is required to elucidate other potential intracellular targets of HPF associated with iron homeostasis and/or lipid peroxidation. Specifically, certain enzymes involved in both the coenzyme Q biosynthesis and the metabolism of mono- and poly-unsaturated fatty acids may play a significant role;
  3. our experiments were exclusively conducted in vitro, which may not fully reflect the complex interactions and dynamics present in an in vivo setting;
  4. during tumor progression, melanoma cells generate neurotransmitters that regulate TME cells and impact tumor homeostasis [70]. Given that HPF can modulate multiple neurotransmitters signaling pathways, it is crucial to conduct in vitro and in vivo studies to further comprehend this complex interplay."

In addition, we have thoroughly revised the English style of the entire manuscript to enhance readability and clarity. Furthermore, we repeated the immunoblots for uPAR and CD133 in FO-1 cells and cyclin D1 in SK-Mel-28 cells, as requested by the other reviewer.

Reviewer 2 Report

This study demonstrates that hyperforin exerts the anti-melanoma effects through pleiotropic effects including the induction of HO-1, triggering lipid peroxidation, disrupting iron homeostasis, and attenuating melanoma progression markers.

The uPAR expression results in FO-1 cells may be replaced with more clear image of the immunoblots in Figure 4.  

Please confirm whether the way of the cell count such as 40 x 10^3 cells is okay or not in lines 163, 171, and 183. 

Author Response

Answers to Reviewer 2

We thank the reviewer for the valuable suggestions he/she addressed in order to improve the present manuscript.

The uPAR expression results in FO-1 cells may be replaced with more clear image of the immunoblots in Figure 4.

Answer

We agree with the reviewer that the image of uPAR expression in the original manuscript had a high background. We repeated the experiment and obtained a new image with improved clarity.

Additionally, we conducted new experiments also to re-evaluate CD133 expression in FO-1 cells and cyclin D1 expression in SK-Mel-28 cells. The new images from these experiments contribute to enhancing the interpretation of the data.

Please confirm whether the way of the cell count such as 40 x 10^3 cells is okay or not in lines 163, 171, and 183. 

Answer

As mentioned in the previous version of the manuscript, 12 x 103 cells were used for transfection (line 163 in the previous version, line 234 in the revised version). For the cytofluorimetric analysis, 40 x 103 cells were used (line 171 in the previous version, line 243 in the revised version). Additionally, 80 x 103 cells were seeded in 24-well plates to protein expression analysis (line 183 in the previous version, line 254 in the revised version). 

In addition, we have thoroughly revised the English style of the entire manuscript to enhance readability and clarity.

We also added at the end of the Conclusion Section certain limitations of our work, as suggested by the reviewer 1

Round 2

Reviewer 1 Report

The authors adequately revised the manuscript, and it should be of strong interest to the readers